# Capturing the Spectrum of Social Media Conflict: A Novel Multi-objective Classification Model

## ABSTRACT

Social media has emerged as a widespread phenomenon, with numerous users engaging in observing, creating, and distributing content. The growing content has led to user conflicts, encompassing bullying, aggression, harassment, and threats. Consequently, recent research has aimed to identify and address these openly hostile forms of social conflict. However, the less overtly hostile yet equally damaging types of conflict, including teasing, criticism, and sarcasm, have been overlooked in current studies.

Our aim is to detect these subtle forms of conflict, while also including openly hostile forms, by developing a novel multi-objective classification model. This innovative approach leverages class based reward functions to improve model performance. Reward functions serve as potent signals capable of mitigating the intricacies of misclassification in multi-class scenarios. By incorporating various rewards within the model architecture, harnessing the power of a decision transformer, we achieved significant improvements in classification performance. Our experiments on three datasets demonstrate superior recall, precision, f1-score, and accuracy compared to traditional state-of-the-art deep learning classifiers. Furthermore, we analyse class ambiguity and its impact on model performance as well as conducting thematic analysis on model misclassifications. We will share the code and datasets at github.com/anonymous.

## KEYWORDS

Deep Learning, Social Media Hate, Social Media Conflicts, Decision Transformer Framework

**ACM Reference Format:**
. 2018. Capturing the Spectrum of Social Media Conflict: A Novel Multi-objective Classification Model. In *Proceedings of* . ACM, 11 pages. https://doi.org/XXXXXXX.XXXXXXX

## 1 INTRODUCTION

Social media has evolved into the primary avenue for interpersonal communication, with extensive research and evidence supporting its widespread usage within society [5, 56, 16]. Social media users disseminate their opinions, views, and thoughts to a broad, global audience. Consequently, increasing negative behaviours and interactions on social media platforms have been observed [2, 63, 8]. Several factors contribute to the escalation of these detrimental behaviours encompassing; anonymity, the absence of tangible physical and social cues, and the absence of accountability for users who propagate harmful content [33]. All of these factors lead to a toxic online environment, with severe consequences to the well-being of individuals and communities[1].

Existing research methodologies aim to discern overtly hostile and socially damaging negative actions, such as hate speech, aggression, and cyberbullying, primarily due to their evident nature [28, 4, 57]. Nevertheless, it has been shown that more subtle negative behaviours, encompassing sarcasm and teasing, wield a significant detrimental influence on user well-being [10, 41, 70, 45], as their perceived harm varies subjectively. We contend that lighter conflicts, such as criticism, sarcasm, and teasing, form part of a spectrum of negative behaviours. This spectrum includes more extreme behaviours like harassment and threats, which have received more extensive research attention. For instance, all depicted user behaviours in Poletto et al. [57], are situated at the end of the spectrum. The lesser behaviours, though not directly overlapping, as slightly intersecting with toxic and occasionally offensive behaviour, primarily existing in an adjacent category. Nonetheless, no research focuses on detecting these broader manifestations of conflicts, and hence it becomes imperative to investigate a diverse spectrum of adverse user interaction behaviours on social media, collectively referred to as "conflicts".

With the increasing recognition of conflicts as a significant research area and a growing social issue [28], there has been a rise in the availability of datasets and models focused on the phenomenon [50, 1]. Existing research has centered on binary hate datasets and published papers, with fewer multi-label and multi-class datasets available [23, 27, 52]. While these resources have contributed valuable insights, it is essential to note that current research datasets have limitations and do not encompass a wide range of social media conflicts. Hence, our work utilises a multi-class conflict dataset, developed exploiting hate netnographies [43], which has not yet been used in multi-class classification tasks.

In multi-class classification, a substantial challenge arises due to the inherent interaction between distinct classes, leading to heightened complexities in distinguishing and isolating specific class instances. Particularly within multi-class scenarios, the occurrence of cross-talk between classes introduces a significant hurdle, consequently impacting the effectiveness of classification [44]. This challenge is worsened in social media scenarios due to the ambiguity, noisy and error-prone constitution of social media data [11]. In addition, the conventional strategy of constructing test collections from social media, especially exploiting distant supervision, contributes to this situation's intricacies. In hate classification research, despite other researchers identifying the complexity and ambiguity of defining these behaviours, there has been no discussion on the profound impact of these issues on classification performance.

Due to the lack of this type of research, we investigate detecting conflicts effectively and experiment with social media data in

---

[1]https://www.rtor.org/2019/10/07/ways-a-toxic-environment-can-be-detrimental-to-your-mental-health/

prominent consumer brand communities. Brand communities are pages, groups, and timelines controlled by brands which provide the opportunity for consumers to interact with the brand and each other [15]. These communities embrace substantial global user cohorts, thereby rendering them of notable significance for consumer conflict research. Prior work within the marketing domain has underscored the challenges brands encounter in relation to conflicts. These challenges include but are not limited to; serious damage to brand reputation, harm to consumers well-being, withdrawal of users from the brand community, and consumer purchase intentions [14, 72]. Therefore, it is essential to undertake a more comprehensive exploration of these conflicts. In addition, we experiment with two other publicly available datasets to demonstrate the generalisability of our approach to existing tasks within the domain alongside the wider range of conflicts we propose here.

As discussed, current approaches fail to capture the spectrum of conflicts, focusing research upon methods which detect only the extreme forms of hate. Therefore, in this paper, we propose a novel approach that involves framing the classification problem as a supervised learning task utilizing class based rewards within the classification model to encourage specific model behaviour. To this end, we leverage the Decision Transformer Model [19], which has exhibited success in domains involving vision and language modelling [46, 37]. However, its application in the realm of text classification remains largely uncharted. The Decision Transformer model introduces a pioneering strategy for transforming reinforcement learning problems into sequential decision-making problems by utilising the Transformer architecture [67]. Nevertheless, transitioning reinforcement learning models to classification scenarios is more complex. Hence, we devise an innovative approach by exploiting the reward functionality aspect of the Decision Transformer framework, introducing a novel class-based reward computation mechanism. The focal point of this reward function is to optimize class distances with the overarching goal of enhancing classification performance.

We make the following contributions in this paper:

- We formulated multi-class text classification as a supervised learning problem incorporating novel reward functionality and developed an effective end-to-end classifier, called conflictDT.
- We investigate the role of reward functions to encourage specific classification behaviours.
- We evaluated this approach in three datasets using multiple metrics and with statistical validation. One of these is a novel multi-class conflict dataset containing a spectrum of negative social media behaviours.
- We analyse classifier performance beyond traditional metrics, conducting a popular social science technique, thematic analysis, to investigate misclassification trends.

## 2 RELATED WORK

There are a multitude of works investigating hate detection that have developed various machine-learning models and explored different classification algorithms and feature representations. Advancements in deep learning have increased the effectiveness significantly [51]. Deep learning algorithms, including BERT [24],

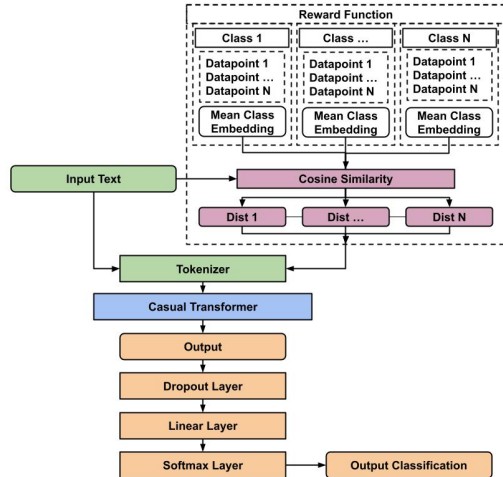

**Figure 1: Model Diagram of ConflictDT.**

LSTM [35], and CNN [55], have demonstrated impressive capabilities across different text classification tasks. Khan et al. [39] proposed a CNN-based framework called HateClassify for labelling social media content, achieving competitive multiclass accuracy. Researchers have explored techniques to enhance the performance of these algorithms further [21, 54]. Ali et al. [3] developed a LSTM-GRU model, combining deep learning and graph analytics, which exhibits state of the art performance over a six class hate speech dataset.

The Transformer architecture [67], a highly effective generative sequential encoder commonly utilized in language modelling and recommendation tasks [47, 66], has revolutionised text classification tasks. This is attributed to its capacity to capture relational information using self-attention mechanisms. Salminen et al.[61] focused on detecting hate across multiple platforms and found models with BERT features superior. Geet D'Sa et al. [21] conducted hate speech classification using word embeddings and Deep Neural Networks (DNN), with the fine-tuning approach using BERT achieving significant improvements. Caselli et al. [17] introduced HateBERT, a retrained BERT model that outperformed the base BERT model in detecting abusive language.

## 3 METHODOLOGY

**Problem Statement:** We first define the problem as a text classification task with a dataset $D = (x_1, y_1), (x_2, y_2), ..., (x_n, y_n)$ where each datapoint contains a text $x$ and a corresponding true label $y$. The discrete action $y$ taken by the model is the classification of the data point into the set of classes. Each true label $y$ belongs to a finite set of classes $C = 1, ..., N$ where N is the number of classes.

### 3.1 ConflictDT classifier:

The core idea of our methodology is to improve upon classification performance when solely using raw datapoint text. Due to class ambiguities, nuanced behavioural multi-class classification often leads to erroneous decisions. We have developed an end-to-end

classifier which uses additional signals alongside text when classifying datapoints, thus improving classification performance. Instead of relying solely on the raw datapoint text input we also calculate numerical features, for example the measurement of the separation between various classes and the text input. These signals are then combined and tokenized before being passed to a fundamental transformer model (e.g., BERT).

To integrate the varied signals, we harness the capabilities of the Decision Transformer Framework [19]. The decision transformer adeptly consolidates the information derived from the assorted rewards, facilitating a comprehensive analysis and well-informed decisions regarding classification. This approach successfully navigates the intricacies present within multi-class classification tasks, adeptly accommodating the subtle distinctions and resemblances among the different classes. A visual representation of our proposed architectural framework is presented in Figure 1. Given a piece of text, we compute the distances between the text and each class. We then combine these singals and the raw datapoint text using the decision transformer framework. The decision transformer framework uses a base classifier (say BERT), to produce an output from the combined input. As shown in Fig. 1 (Causal transformer), the decision Transformer layer takes the stacked input of text and reward features and produces a classification output. The output of the decision transformer layer then subsequently passes through several linear transformations before being classified. Ultimately, the culmination of the process yields a classification output determined by a confluence of datapoint text and rewards. Gontier et al.[32], utilise the decision transformer framework within a text-based game setting to train a reinforcement learning model. They incorporate additional signals computed from the input data with natural text elements of the game, using the decision transformer framework. We adopt a similar approach to incorporate our class based distance rewards with the text datapoints from the datasets.

The model follows the standard architecture of a deep learning transformer model, shown in Equation 1, with the exception of the input $x$ which would normally be a textual embedding and is instead a combined text and reward feature embedding.

$$z = \sum_{i=1}^{n} w_i * x_i + b$$
$$a = \psi(z) \tag{1}$$

Transformer model; where w, weight, and b, bias, are trainable parameters. $\psi$ is the Softmax activation function. Action, a, is selecting one of the N dataset classes.

As depicted in Figure 1, the text and reward features are combined before going through the tokenizer and decision transformer layer, which further goes through a number of transformations. The output of the transformer model is first passed through a dropout layer to help prevent overfitting. The dropout layer randomly sets a number of the input tensor elements to zero depending on a set probability. Next a linear layer, shown in Equation 2, is used to produce the output logits, a vector of raw predictions for each class within the task.

$$y = xA^T + b \tag{2}$$

Linear layer equation; where x is the input, A is the weight, b is the bias, and y is the output.

Finally, a softmax layer is then used to output the predicted class for the datapoint. The softmax function takes as input the values from the dropout layer and turns them into a probability distribution. The class with the highest probability within the softmax layer output is chosen as the predicted class.

$$softmax(Z)_i = \frac{e^{Z_i}}{\sum_{j=1}^{N} e^{Z_j}} \tag{3}$$

Softmax layer equation; where Z is the output vector from the previous layers, $Z_i$ is the i-th element of Z, the value of e = (2.718), and N is the number of classes.

In order to remove cross-talk between classes, we design a reward modelling scheme. Primarily, we harness rewards as metrics of distances between the task classes and the current text embedding. As shown in Figure 1, the input text is combined with reward features and then undergoes an embedding procedure (e.g., BERT), followed by the transformer model's output classification. Although the Decision Transformer paper [19], which formed the foundation for this work, utilised the GPT-2 model [59], we ultimately decided to use the BERT model [24] as the underlying transformer model within our research. This was due to BERT's superior performance in initial experiments, a decision reinforced by the findings of the experiments within this paper (see Tables 4 and 5). Rewards can be modelled by various techniques tailored to suit distinct problem tasks, and we have experimented with different schemes, emphasizing the diversity of our approach.

**Reward Function:** The reward functions are instantiated as set of distances between mean class embeddings, formed from class sets, and the embedding of the text to be classified. These distances direct the classifier in making the correct decision. We explored numerous variations of reward functions based on distances during our experiments (section 4). In order to calculate the reward function, the cosine similarities between the text input embedding $e_i$ and each mean class embedding $\bar{y}$ are calculated as shown in equation 5 and the dashed line box of figure 1. Next these cosine similarities are normalised and subsequently scaled from 1 to 100. We took this decision following research published by Wallace et al. [69], who investigate various NLP model's understanding of numbers. They find that transformer models do capture numerical features but some, notably BERT, struggle with decimal numbers. A combination of these similarities is then returned as the reward, this then consequently forms part of the model input.

We adopted the cosine similarity metric for our investigation to compute class distances, exploring two distinct methods. The first method entailed calculating the mean vector for each of the class clusters (equation 2) and gauging the cosine similarities between these mean vectors and text embedding. Conversely, the second method involved computing the cosine similarity between each

possible pair of text embeddings of given text and elements of each class, then calculating an average cosine similarity. Upon evaluation, the first approach was deemed more efficient, thus becoming the chosen method implemented within our model.

As a basis for the reward function mean text embeddings $\bar{y}$ for each class have to be calculated, these are then used in the reward function as described below. The mean text embeddings are calculated as follows where $y_i$ are individual text embeddings within each class.

$$\bar{y} = \frac{1}{N} \sum_{i=1}^{N} y_i \tag{4}$$

$$r_{\bar{y}e_i} = Cos\,Sim(\bar{y}, e_i) \tag{5}$$

Loss for the model is calculated using cross-entropy loss as follows:

$$L_{CE} = -\sum_{i=1}^{N} t_i log(p_i) \tag{6}$$

for N classes in the dataset, where $t_i$ is the true label and $p_i$ is the softmax probability for the, $i^{th}$ class.

## 3.2 Multi-class Conflict Dataset

The prevailing datasets in the realm of conflict classification predominantly concentrate on overtly antagonistic manifestations of adverse interactions [23, 39, 31], thus disregarding the complete spectrum of conflict. This constraint impedes our comprehension of these interactions' repercussions on users, particularly the potential harm inflicted on vulnerable users such as children. It becomes imperative to possess the means and mechanisms for identifying more expansive variants of conflict. In our research, we employ a more all-encompassing dataset of Facebook comments, affording an initial opportunity to scrutinize more nuanced manifestations of conflict.

This conflict dataset was compiled by applying hate netnographies [42]. These netnographies facilitated the identification of six discrete categories of conflicts. These categories were deduced from a sixteen-month netnographic investigation across four online brand communities [14]. Throughout this investigative period, the researchers meticulously examined and classified many consumer conflicts, aiming to comprehend the diverse manifestations of such conflicts. The researchers adopted a dual-coding methodology to uphold the integrity of the annotation procedure, involving the active participation of two social science researchers. The initial phase encompassed the first researcher's deductive identification of incidents involving consumer conflicts. An independent analysis of the data was then undertaken by a second researcher. Both researchers subsequently engaged in a comprehensive assessment of the reliability and applicability of their respective analyses. This deliberation was accompanied by extensive discussions to resolve any divergence in their interpretations. This rigorous and meticulous process culminated in identifying and classifying six distinct categories of conflicts, namely 'Teasing', 'Criticism', 'Sarcasm', 'Trolling', 'Harassment', and 'Threats'. Final conflict dataset statistics shown in Table 1.

**Table 1: Table showing conflict dataset class sizes**

| Lesser Conflicts | | More Extreme Conflicts | |
|---|---|---|---|
| Class | Datapoints | Class | Datapoints |
| Teasing | 208 | Trolling | 1089 |
| Criticism | 698 | Harassment | 1098 |
| Sarcasm | 577 | Threats | 482 |

## 4 IMPLEMENTATION AND BASELINES

### 4.1 Implementation

All models were trained over four epochs with a learning rate of 2e-5 and the same Cross Entropy Loss Function from Pytorch, *torch.nn.CrossEntropyLoss* [20]. The Dropout [25], Linear [48], and Softmax [64] layers also use the corresponding modules from the Pytorch library. For the Dropout Layer we set p=0.2. For the Linear layer, the *nn.Linear* module takes two parameters; *in_features*, the number of input features, and *out_features*, the number of output features. For a classification task the number of output features is equal to the number of classes. We optimise models using AdamW [49]. For the baseline models, all hyperparameters were used from the published papers, trained using the same setup and fine-tuned on the same training, validation, and test splits from each dataset. The train, validation, test splits for all datasets were 80%, 10%, and 10% respectively.

### 4.2 Metrics and Statistical Tests

In order to evaluate classifier performance we used Accuracy, F-1 Score, Recall, and Precision. Due to the imbalanced nature of the conflict dataset, it is important to evaluate using both F1-Score and Accuracy. In order to validate the significance of our results we conducted statistical testing for the model classification performance. The statistical test we chose is the paired two-tailed t test. To get the performance metrics for each model we performed 5 K-fold cross-validation and took the average results. For each dataset in experiment two we calculated the t value between the Decision Transformer model and the base BERT model. The t-value was then used to calculate a p-value which was evaluated against alphas of $\alpha = 0.05$ and $\alpha = 0.10$.

### 4.3 Baselines

The following baselines were used:

- BERT: due to proven performance in classification tasks, especially within social media text problems [71, 34]. For the BERT model, we used the BERT-Base uncased pre-trained model with 12 layers, 12 heads, 768 hidden size, and 110M parameters.
- Flan-T5: A state of the art generative language model which can be fine tuned for text classification. Edwards and Camacho-Collados evaluate Flan-T5 for a multitude of text classification problems, showing strong performance metrics.[26]. We use the Flan-T5 base model with 248M parameters uploaded to the Huggingface repository by the google team [36].

- GPT-2 : demonstrated effectiveness in text classification tasks and use in Decision Transformer work by Chen et al. [51, 6, 19]. For the GPT-2 model, we used the default GPT-2 model with 12 layers, 12 heads, 768 hidden size, and 117M parameters.
- HateBERT: [17], focusing on abusive language detection, specifically offensive, abusive and hateful language. Hate-BERT features intensive pre-training on social media comments before being deployed for fine-tuning domain-specific tasks. For HateBERT we used the default model provided by the authors with 12 layers, 12 heads, 768 hidden size, and 110M parameters.
- DistilBERT: a lightweight variation of the base BERT model, has been proven to be an excellent competitor to the traditional BERT model, with Sanh et al. [62] stating; "it is possible to reduce the size of a BERT model by 40%, while retaining 97% of its language understanding capabilities and being 60% faster". We elected to benchmark DistilBERT for its successful use by Mutanga et al. [53] within a hate classification task. For the DistilBERT model we used the default DistilBERT model with 6 layers, 768 hidden size, 12 heads, and 66M parameters.

## 4.4 Experiments

*4.4.1 Datasets.* We use the Davidson et al. dataset [23] with 24,000 texts across 'Normal', 'Offensive Language', and 'Hate Speech' classes, the Founta et al. dataset [31] with 44,911 texts across 'Spam', 'Abusive', 'Hateful', and 'Normal' classes and the conflict dataset [14], (Table 1) with 4,052 texts across 'Teasing', 'Sarcasm', 'Criticism', 'Harassment', 'Trolling', and 'Threat' classes.

*4.4.2 Research Questions.*
- R.Q.1 How robust is the multi-class conflict dataset?
- R.Q.2 How effective is the model for multi-class classification? Does the conflictDT Classification Model outperform other baseline models across hate and conflict datasets?
- R.Q.3 Can the reward function alter the model's behaviour and can it be used to improve classification performance?
- R.Q.4 Can we identify common patterns and trends within model misclassifications? Do these trends link with existing social science theories surrounding online communication behaviours?

*4.4.3 Experiment One,* to answer R.Q.1, involves analyzing the conflict dataset, and focusing on how different class characteristics could influence classification performance. This analysis included examining class size, textual features, and calculating class similarity via cosine similarity, as outlined in the methodology. This analytical review aimed to identify trends and features within the dataset that could aid in the design of effective classification systems. Moreover, when introducing a novel dataset to the wider research community, it is crucial to recognize its strengths and weaknesses.

*4.4.4 Experiment Two,* answering R.Q.2, tests the performance of the conflictDT model against other state-of-the-art models. To ensure a comprehensive evaluation of our model, we conducted an in-depth analysis across various popular datasets within the

Hate and Conflict domain. Several researchers have emphasised the importance of model generalisation across datasets [30, 7]. This was particularly relevant for our study due to the relatively small size of the conflict dataset. By testing our model on additional datasets, we demonstrated its ability to generalize and showcased the validity and robustness of it's performance.

*4.4.5 Experiment Three,* investigating R.Q.3, focused on studying the reward function within the conflictDT model using the conflict dataset. The main challenge in classifying social media data sets is class crosstalk, notably exacerbated due to shared properties among classes for example, sarcasm, teasing, and trolling. Our approach in selecting individual reward functions involved a thorough analysis of distinct properties within the conflict dataset, specifically addressing existing misclassification patterns. We conducted a series of tests with multiple reward functions, the goal was to examine the impact of the reward function on overall classification performance. Showcasing the novelty and flexibility of the reward functionality within the model, which allows the prioritization of different behaviours and aspects within the classification problem. In order to further analyse the performance of the BERT and ConflictDT models we also include a breakdown of class performance within the conflict dataset. The reward function was varied while keeping all other parameters constant. We tested five different reward functions: (i) distance between the text embedding and all classes (equation 2); (ii) the distance between the text embedding and lesser and more extreme forms of conflict; (iii) distance between the 'Harassment' class and the text embedding; (iv) the distance between the text embedding and all classes with sequential functionality; (v) no reward with sequential functionality. These reward functions were selected based on our knowledge of the dataset and analysis conducted in the experiments one and two. The first reward function, 'distance between all classes', aimed to prioritize separating all classes within the dataset and could be applied to any classification task. The second reward function, 'distance between lesser and extreme conflict groups', aimed to exploit common characteristics within these two groups, as identified during dataset analysis. The 'Harassment' based reward function was derived from the initial results of our model, which indicated that the 'Harassment' class datapoints were being frequently misclassified into other classes and vice versa (Fig 3). Consequently, we sought to demonstrate the model's ability to prioritize different classification behaviours by designing a reward function to address this trend. The final two variations of conflictDT feature sequential modelling. The Decision Transformer paper that inspired this work [19] utilise transformer models as part of sequence modelling problems. We sought to map this same framework to text classification by mapping states to sentences within the datapoints. We divided the datapoints into sentences, with each state consisting of the previous states plus the next sentence. So the first state would contain the first sentence, the second would contain the first two sentences; and so on. The input for these sequence modelling classification models would then be a combination of the current state text, the output of the reward function for that text state, and the previous classification taken by the model. The model would then provide a classification for each state, with the final state being the entire text datapoint. This sequence modelling approach aimed to exploit the sequential

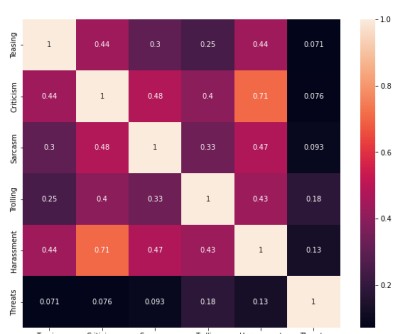

**Figure 2: Heatmap showing class similarity.**

**Table 2: Table showing mean class characteristics within the conflict dataset**

|            | Chars | Words | % Stop Words | No. Sentences |
|------------|-------|-------|--------------|---------------|
| Teasing    | 78.5  | 14.6  | 0.31         | 1.6           |
| Criticism  | 232.9 | 42.7  | 0.41         | 2.9           |
| Sarcasm    | 58.9  | 10.7  | 0.32         | 1.1           |
| Trolling   | 105.4 | 18.6  | 0.32         | 1.6           |
| Harassment | 130.2 | 23.9  | 0.35         | 2.2           |
| Threats    | 275.7 | 50.6  | 0.31         | 5.5           |

nature of text, slowly exposing the model to increased contextual information at each timestep.

*4.4.6 Experiment Four,* answering R.Q.4, involves qualitative analysis of the conflictDT classifier results on the six class conflict dataset. In order to gain an insight into common themes of misclassification we conduct thematic analysis, a popular and recognised technique within social sciences used to complement quantiative approaches, formally defined by Braun and Clarke [13]. Thematic analysis is a method used to make sense of human communication content, in our case online comments, to identify themes or patterns that emerge. We follow the standards set out by Braun and Clark [12] which feature 6 steps; Familiarization of data, Generation of codes, Combining codes into themes, Reviewing themes, Determine significance of themes, and Reporting of Findings. We follow an inductive coding approach where one coder annotates the data, then a second coder reviews the first coders identified codes, patterns, and themes. The two researchers then enter a discussion before finalising the codes, patterns, and themes. The analysis concludes by reporting the resulting themes, definitions, descriptions, and sample comments in order for others to replicate and understand the results.

# 5 RESULTS AND DISCUSSION

## 5.1 Experiment One - Analysing Multi-Class Conflict Dataset Characteristics and Robustness

The main contribution of experiment one is analyzing the novel conflict dataset encompassing a range of negative behaviours. As highlighted by Bianchi et al. [9], there is a scarcity of research on nuanced hate and conflict, thus, this paper fills an evident gap in existing work. The conflict dataset is grounded in established marketing theory, employing social science researchers with expertise in hate and conflict to annotate the data.

Due to our task's complex and real-life nature, we investigated similarity between the six classes using a heatmap, Figure 2, and observed an average similarity of (0.32) between classes. Whilst similarities between some conflicts are low e.g. 'Threats' and the other classes, there is less distinction between others such as 'Criticism' and 'Harassment' which had a high level of similarity (0.71). This similarity poses a challenge for classification, particularly given the small dataset. Higher similarity between classes results in greater difficulty for classifiers as the classification problem is more ambiguous. Brevity of data is another aspect to consider in this context, as such we conducted an analysis of textual features (Table 2). With the character length and number of words means across all classes being just 147 characters and 26.85 words respectively. Although research has established correlation between shorter texts and reduced performance [65], the conflict dataset's short length accurately reflects the domain's characteristics and the classification task. Nonetheless, developing robust methods to effectively handle short texts and overlap between classes is crucial.

By generating a class similarity matrix we delve into the conflict dataset composition beyond class size, gaining insights which further enrich dataset evaluation and development of our custom reward function. For instance, we can observe that, on average, milder conflicts exhibit greater similarities than severe conflicts. Furthermore, we can identify a significant level of similarity between the 'Criticism' and 'Harassment' classes. This could be due to links between the behaviours, previous works [40, 38] have specifically investigated crossover between the two; how actors within interactions can have different perceptions of criticism and harassment, and how criticism develops into harassment. These findings contribute to shaping our reward functions in the context of experiment three. However, despite the merits, the dataset does have limitations. While dataset size is comparable to other works [62], it remains relatively small compared to other extensive datasets available [23, 68]. However, SOTA deep learning models have demonstrated impressive performance working with small datasets [62, 17]. In accordance with limited data availability, the conflict dataset used in this paper is also relatively small. Another drawback is class imbalance; due to the nature of social media data, not all classes were equally represented.

## 5.2 Experiment Two - Comparing Model Performance Across Three Conflict and Hate Datasets

As can be seen in Table 4, the conflictDT model, with a reward function of distance between all classes, outperforms other models in F1 score on the Founta dataset [31]. We achieve superior performance to BERT in F1 score by 1% , whilst also significantly outperforming GPT-2 by 5% in F1 score. The best performing model

**Table 3: Table showing the themes identified in one test set's misclassifications**

| Theme % Misclassified Comments | Definition | Description | Examples |
|---|---|---|---|
| **Linguistic Fluidity** **36%** | A misclassification that occurs due to the lack of definitional boundaries that are inherent to the interpretation of language. | A commonly known phenomenon in linguistics is that of multiple meanings to the same sentence, where meaning interpretation depends on a multitude of unpredictable factors(e.g. one's mood, need for politeness etc; ) Classes are not always clear cut and often have fluid boundaries. Datapoints can contain behaviour which could belong to more than one class. Therefore neither classifier or human can get it totally accurate. | " Quickly! Let's spend our precious weekend time arguing about fast food on the Internet!" "everyone in every country has a right to their own opinion. I'm not sure why it matters what country, it's their opinions and their beliefs. No one is asking you to agree but respect the fact they are entitled to their opinions and beliefs." |
| **Context Dependency** **28%** | A misclassification that occurs when the meaning of a message changes when taken out of context. | Context dependency is a common issue identified by language researchers when studying offensive/impolite communication. The type of behaviour exhibited in a datapoint depends on the context of the prior comments. When classifiers are utilising single comment datapoints this context is not present, as opposed to the human coders who had access to the whole conversation when conducting netnographies. | "Picture that says: it's a joke not a dick, don't take it so hard." "If Andy Dwyer wasn't likable." |
| **Humour Ambiguity** **20%** | A misclassification that occurs when a message fails to convey that it was meant in humor and/ or was good vs bad-natured. | Linguists have long recognised the lack of clarity inherent to humor as a quality on which humor often relies. Humour has been recognised as a particularly challenging area of NLP. Humor can be taken two ways and is subjective meaning the dominant type of humour behaviour is often ambiguous within datapoints. | "The Not So Fresh Princess of Stale Air." "Keep checking on Arsenal every champions league game mate ! Mu won't be there! Arsenal are here to entertain! unlike your confused club!" |

**Table 4: Table showing the model performances across datasets. Best results in bold, second best in italics. * Denotes a significant improvement over the baseline BERT model at $\alpha = 0.1$, *p*-values are in parentheses.**

|  | Founta et al. | | | | Davidson et al. | | | | Conflict Dataset | | | |
|---|---|---|---|---|---|---|---|---|---|---|---|---|
|  | Acc | F-1 | Rec | Pre | Acc | F-1 | Rec | Pre | Acc | F-1 | Rec | Pre |
| BERT | *0.77* | *0.76* | *0.77* | **0.78** | 0.87 | *0.86* | *0.86* | 0.86 | *0.69* | *0.61* | *0.61* | **0.65** |
| HateBERT | **0.78** | 0.73 | 0.73 | 0.73 | 0.86 | 0.86 | *0.86* | 0.87 | 0.55 | 0.52 | 0.52 | 0.52 |
| DistilBERT | *0.77* | 0.69 | 0.67 | 0.70 | 0.85 | 0.86 | *0.86* | 0.86 | 0.55 | 0.53 | 0.52 | 0.54 |
| GPT-2 | *0.77* | 0.72 | 0.73 | 0.72 | **0.90** | 0.73 | 0.72 | 0.73 | 0.59 | 0.51 | 0.51 | 0.55 |
| Flan-T5 | *0.77* | *0.76* | 0.76 | *0.77* | 0.87 | *0.87* | 0.86 | **0.88** | 0.67 | 0.60 | *0.61* | 0.61 |
| conflictDT | *0.77* | **0.77** | **0.78** | **0.78** | *0.89* | **0.88*** **(0.07)** | **0.88** | **0.88** | **0.71** | **0.63*** **(0.08)** | **0.64** | *0.62* |

for accuracy was HateBERT with 0.78, although importantly conflictDT outperformed HateBERT in F1 score by 4% and matched all other model's performance in accuracy at 0.77. On the Davidson et al. dataset[23], it outperforms all other variations of BERT by 2% in F1 score. However, it only marginally outperforms Flan-T5 by 1% in F1 score. GPT-2 outperforms conflictDT by 1% in accuracy although this is countered by significantly worse performance in

F1 score, where conflictDT outperforms GPT-2 by 15%. Within the conflict dataset we see the conflictDT model outperforming the next best model, BERT, by 2% in F1-score and accuracy.

To test statistical significance, we performed paired t tests between conflictDT model and the base BERT model. In each test there were 4 degrees of freedom. For the conflict dataset, the *t*-value is 1.53 and the *p*-value is .08. For the Davidson dataset, the

**Table 5: The effects of reward functions in the conflictDT model. Best results in bold, second best in italics.**

|                              | Acc  | F-1  | Rec  | Pre  |
|------------------------------|------|------|------|------|
| BERT                         | *0.69* | *0.61* | 0.61 | **0.65** |
| GPT-2                        | 0.59 | 0.51 | 0.51 | 0.55 |
| Flan-T5                      | 0.67 | 0.60 | 0.61 | 0.61 |
| Dist all Classes             | **0.71** | **0.63** | **0.64** | 0.62 |
| Dist Less & Gtr Class Groups | 0.67 | 0.59 | 0.60 | 0.59 |
| Dist 'Harassment'            | 0.68 | *0.61* | *0.62* | *0.63* |
| Sequential no Reward         | 0.66 | 0.59 | 0.59 | 0.62 |
| Sequential w/ Reward         | 0.68 | 0.60 | 0.60 | 0.62 |

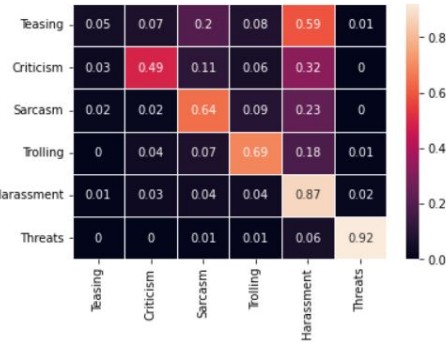

**Figure 3: Heatmap showing BERT model class performance on the conflict dataset.**

$t$-value is 1.62 and the $p$-value is .07. Therefore, we can say that our ConflictDT model significantly outperforms the base BERT model on the Davidson and Conflict Datasets at a threshold of $\alpha = 0.1$ but not at a threshold of $\alpha = 0.05$. For the Founta et al. dataset the t test did not show a significant p value, with a $t$-value of 1.13 and a $p$-value of 0.15. Although this result is not significant using thresholds of either $\alpha = 0.05$ or $\alpha = 0.10$, the ConflictDT was still tested using 5 k-cross fold validation and thus still shows merit in it's performance.

The results across three popular datasets show our conflictDT model performs well within hate and conflict text classification. With regards to the F1 score metric our model outperforms SOTA models on two datasets whilst achieving marginally better performance on the third. By showing that our model produces robust results whilst also generalising across datasets, we increase the reliability of our results. This, therefore, mitigates the shortcomings of the conflict dataset, which suffers from class imbalance and data size. In addition, it also shows the adaptability of the reward functions for new data scenarios.

### 5.3 Experiment Three - Evaluating ConflictDT's Reward Functionality Performance and it's effect on individual class performance.

Experiment three delved into a comprehensive examination of reward functionality in our conflictDT model. Table 5 shows the effects of different reward functions within the model which produced a variety of classification performance scores. Here we explore the potential of various reward functions in directing the classifier towards correct decision. All variations of reward function in the conflictDT model show comparative performance to the BERT model. Rewards of distance between lesser and more extreme hate groups and distance between 'Harassment' and other classes are slightly outperformed by BERT in accuracy. The 'Harassment' reward matches BERT in F1-score whilst the 'Lesser' and 'Greater' group reward sees a decrease of 0.02 in F1-score. The conflictDT model with a reward of distances to all classes outperforms BERT by 4% and 2% in F1-score and accuracy. As in experiment two, models based on BERT outperform GPT-2. The best-performing model was the conflictDT model with the reward function of distance between the text embedding and all classes. Both the sequential versions of the model fail to make significant improvements on the base model. We theorise that this may be due to the short text length

within the conflict dataset, quantified in Experiment One Table 2. The additional data gained at each timestep may simply not be enough to aid the model, and instead causes confusion within the signals.

Examining the heatmaps for the BERT and ConflictDT models over the six class conflict dataset, we observe that ConflictDT obtains higher true positives in Teasing, Criticism, and Trolling. The same true positives in Sarcasm, and lower true positives in Harassment. Additionally, the ConflictDT model results in a reduction in all other classes being misclassified as Harassment, while Harassment is more frequently misclassified into Sarcasm and Trolling.

The key finding of this experiment is that the reward functionality empowers us to modify model behaviour. This distinctive capability to leverage prior knowledge and counteract specific trends in model classification is a significant advantage. Both general rewards, such as the distance between all classes, and rewards that target specific high misclassifications, can lead to improved model performance. However, it should be noted that specific rewards, such as the distance between less severe and more extreme classes, can result in decreased performance. Considering the earlier dataset analysis, this outcome was unexpected as this reward function was anticipated to enhance performance. We had theorised that promoting distance between the two class groups would reduce misclassification between them. This decrease in performance may be due to the ambiguity of classes within each group or the differences between the groups not being significant enough.

### 5.4 Experiment Four- Thematic Analysis of Misclassified Comments

The thematic analysis employed produced three main misclassification themes which we have called Linguistic Fluidity, Context Dependency, and Humour Ambiguity. These themes alongside their definitions, descriptions, examples, and frequency can be seen in table 3. The theme of linguistic fluidity encompasses the fluid or blurred boundaries between class behaviours. Although datapoints tend to have a dominant behaviour, they can contain aspects of multiple class behaviours. This presence of ambiguous behaviours

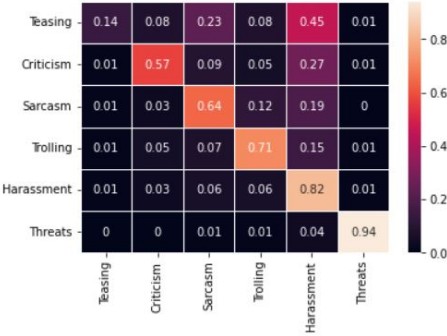

**Figure 4: Heatmap showing ConflictDT model class performance on the conflict dataset.**

has been identified in other works. For example, both Jhaver et al. and Kim et al. [38, 40] identify crossover between Criticism and Harassment. They discuss how Criticism can develop into Harassment and how often the true identify of the behaviour is subjective. Additionally, we can see this theme emerging within hate terminology. Fortuna et al. [29] discuss how terminology differs between research papers within the hate domain, leading to ambiguity between behaviour classes in different datasets and misinterpretation of the identity of behaviours within the hate domain.

The second theme, Context Dependency, relates to those datapoints where the identification of the behaviour class partially or majorly relied on context which was unavailable to the classifier. Whilst single social media datapoints frequently contain obvious negative behaviours they are often contained within or associated with a larger 'chain' or 'thread' of comments. Therefore, the behaviours prescribed to the individual datapoints are related to the other activity within those chains. As this information is unavailable to the classifier it leads to misclassification as the correct behaviour is context dependant. This theme has been identified and attempts made to address it by a plethora of researchers within the classification domain [58, 22, 60, 18], with many seeking to develop models which consider entire social media conversations within classification problems.

The final theme, Humour Ambiguity, relates to the difficulty of NLP models to identify specific forms of humour. Humour has been recognised as a particularly challenging area of NLP. Humour is largely subjective and often relies on subtle cues or inside knowledge. Take for example the first humour ambiguity datapoint in table 3, 'The Not So Fresh Princess of Stale Air.' which belonged to the 'Trolling' class but was misclassified as 'Harassment', the difficulties in identifying this datapoint are two-fold. First the model has to understand that the comment has a humour aspect and is not just an insult, secondly the understanding that the datapoint references a popular comedic show from the 1990's. Whilst a human could reasonably understand that the comment has a humourous aspect, the model struggles with these nuanced factors. We can see the pattern develop in the second example of humour ambiguity where the nuance of humourous teasing is dependent on knowledge of football clubs and the banter which exists between the fans of said clubs.

## 5.5 Future Works

In future work, we will expand the dataset size. While we have taken steps to mitigate the impact on the significance of model performance by testing on other popular datasets, increasing the scale of the conflict dataset would enhance the reliability and robustness of our results. Additionally, there is room for further exploration of reward functions. Our experiments focused on the distance between class text embeddings, but future reward functions could encompass other aspects, such as distance between datapoint text embeddings within classes or similar emotions and sentiments within classes. Lastly, research could focus on the sequential modeling we briefly explored in experiment three, the decision transformer work by Chen et al. [19] showed sequential modelling to achieve great success in other IR tasks. Given the sequential nature of text, the decision transformer framework could be exploited further, perhaps considering alternative representations of states within the text, such as individual words, different groups of words, or even paragraphs. Although we provide a qualitative analysis of our models misclassifications using a popular and well recognised social science technique there is potential to perform a quantitative explainable AI experiment. This would aid in understanding the role that the reward functionality plays within the model. Finally, a case study investigating the use of this model within a simulated real life scenario could be conducted, investigating the feasibility of the model as a tool for use within social media platforms. Such a case study would further demonstrate the potential important impact of a model capable of identifying a range of behaviors whilst highlighting responsible AI practices.

## 6 CONCLUSION

We successfully developed and evaluated a novel social media conflict classification system. The novelty of our research lies in our innovative approach to modeling multi-class classification. Specifically, we have devised a novel reward scheme that extracts nuanced signals essential for addressing the complexities inherent in multi-class classification scenarios. Leveraging the decision transformer architecture, we effectively integrate these classification signals, thereby enhancing the classification process. In addition, our approach is applicable to various problem domains, beyond the scope of hate and conflict analysis. The use of a dataset covering a spectrum of social media conflict, including less extreme forms, fulfils a gap in the current literature which predominantly focuses on extreme forms of hate. A quantitative analysis of this conflict dataset, examining class characteristics, class similarity, and individual class classification performance contributes to the robustness of the work. Our model significantly outperformed state of the art models on two multi-class conflict and hate datasets, and achieved non-significant outperformance on the third. Finally, we conduct a qualitative analysis of model misclassifications, employing thematic analysis to identify trends and patterns within misclassifications. With the successful results achieved, the classification system and custom dataset can serve as the foundation for further exploration into social media conflict.

, ,

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
