# OpenReview forum: "Capturing the Spectrum of Social Media Conflict: A Novel Multi-objective Classification Model"
_ACM.org/SIGIR/ICTIR/2024/Conference — ICTIR 2024_

### Official Review · Reviewer_tHvg · 2024-05-13

**Rating:** 1
**Confidence:** 4

**Objective Part Of Review:**

This paper proposes a novel multi-objective classification model to detect these subtle forms of conflict (e.g., teasing, sarcasm). The proposed approach leverages class based reward functions to improve model performance. Experiments are applied on three datasets.

The problem is stated clearly and the methodology is well described in the paper. The authors use three different datasets in their experiments and they use several baselines to compare their method with. Results are presented from different perspectives.

The main limitation of the paper is that the improvement in the results is little and there is statistical significance at 0.1 and not at a lower threshold. So the question is whether the distances are those that made the difference in the results or the difference is random.
A second limitation is that I found the related section short, I would like to see more related work discussed there.

**Subjective Part Of Review:**

The paper was easy to follow and the problem is important and very relevant. The analysis that the authors do in the results section is thorough and interesting. However, the improvement is not big which is the main limitation of the paper.

---

### Official Review · Reviewer_NFzY · 2024-05-17

**Rating:** 0
**Confidence:** 4

**Objective Part Of Review:**

This submission extends hate speech classification to include less intense forms of negative speech such as teasing. In addition to building a dataset that can be used for this expanded task, it explores a set of classification approaches that incorporate signals that capture the proximity to the six harmful-speech categories as well as functions of them.

The problem is very well described and the goals of the work seem reasonable. The dataset and the experiments are also well described.

Here are some things that seem related to "objective" issues (though the distinction is not clear to me as a reviewer!):

* Introduction, first sentence. I'm sorry, but  social media is not "THE primary avenue for interpersonal communication." It's "A primary avenue" or something. I mean, I'd say that we talk to each other a lot more than we use social media. At least, I hope so.
* I think some sketch of what a netnography is ought to be there. Citations are good, but I a sentence that summarizes the key idea of what it is would be helpful.
 * In section 3.2, when you are describing the annotation and having researches look at things, what was the eventual level of agreement between the researches? That is, what if no one agreed at all? If agreement is poor that can suggest a problem is too difficult, so this is an important data point.
* Having HateBERT as a baseline is good, but presumably it doesn't know about the new classes, so it is necessarily at a disadvantage and isn't a good baselines.
* In 4.4.1 you list datasets. They are not the same as that described in 3.2. Unify those so it's clear what is happening.
* Figure 2. Can you make the text bigger? It's a challenge. What how is this similarity calculated?Is it to the class centroids or something else?
* In 5.2,you talk about an improvement of 1% in F1. Is that useful? It is meaningful? Does it have any impact on anyone?
* Figures 3 and 4. I believe this is comparing how BERT did on the classes compared to the truth data? What is the measure in there? Is the X or the Y axis the truth set? Can you put these two figures together as a single figure so that it's easier to compare them?
* in the conclusion, you state that "our approach is applicable to various problem domains." How do we know this?

**Subjective Part Of Review:**

I like the hypotheses and the presentation of how they work. I think the motivation for the new classification approach (incorporating whether a text roughly matches known examples of the classes) is good.

However, I found the presentation a little bit difficult to follow at times. The heat maps, in particular, did not seem very useful toward understanding things. That may be because in the end, the new approach resulted in modest to low gains in effectiveness, meaning the heat maps were less useful because there was less to show. It may also mean that this submission is trying to solve a problem that is not actual present in the data in the right way.


Here are some very small issues that should be easy to fix or at least easy to decide whether to fix or ignore:

* Introduction. The sentence  starting "The lesser behaviours" is not a sentence: there is no main verb.
* Maybe this is a cultural thing, but normally one uses a colon (:) rather than a semicolon (;) to introduce a list. So in the intro, "behaviours encompassing: anonymity, the absence of..."
* In the contributions, you "formulate" and "evaluate" rather than "formulated" and "evaluated" for parallelism with the others.
* I don't get what DT in conflict DT means. It'd be nice if you tossed in a quick explanation.
* In 3.1, singals $\longrightarrow$ signals
* In 4.2, rather than "5 K-fold" I think you want "5-fold" since the other looks suspiciously (and implausibly) like "5000-fold"
* Also in 4.2, you are suddenly talking about "experiment two" which we have never heard of previously. Needs some explanation.
* In 4.4.5, "work [19] utilise" $\longrightarrow$ "work [19] utilise**s**"
* The 5.3 heading, it's $\longrightarrow$ its
* In 5.3, the paragraph starting "Examining the heatmaps" needs a pointer to Figures 3 and 4.
* In 5.4, I suspect that: true identify $\longrightarrow$ true identity
* In conclusion, "fulfills a gap" $\longrightarrow$ "fills a gap"

---

### Official Review · Reviewer_JZqG · 2024-05-17

**Rating:** 1
**Confidence:** 3

**Objective Part Of Review:**

In this work the authors focus on the problem of identifying hostile forms of social conflict and additionally aim to accurately detect a wide spectrum of conflicts that is not only tailored to extreme types of hate speech. To this end, they propose a multi-objective classification model that incorporates reward functions defined by distances between mean class embeddings. They experimentally evaluate the model under two established datasets and a novel multi-class conflict dataset (containing six distinct categories of conflicts) and showcase their results compared to baseline approaches.

In general, the problem is clearly stated and the paper is well presented. The structure of the work is easy to follow and the proposed approaches are backed by experimental evaluation.

**Subjective Part Of Review:**

Overall, the work presents novel contribution to the area and should be considered for acceptance. The experimental section supports the premise that the proposed model can achieve superior performance compared to baseline approaches. However, that performance increase is somewhat marginal and not statistically significant (alpha = 0.1) under all of examined metrics. The inclusion of the compiled multi-class conflict dataset offers an additional intuitive way of evaluating the model.

Minor comments:
1) The related work section could be noticeably extended (perhaps with additional material related to RQ4)
2) Please capitalize all cross-references to Tables, Figures etc. in the main text.

---

### Official Review · Reviewer_WPug · 2024-05-18

**Rating:** 1
**Confidence:** 3

**Objective Part Of Review:**

The paper utilizes a diverse conflict dataset encompassing a range of social media behaviors, which fills a gap in the current literature that primarily focuses on extreme forms of hate. The authors conduct a thorough analysis, including class ambiguity and thematic analysis of misclassifications.

There are several typos and clarity issues in the paper, such as the missing brackets in the definition of  C . These need to be addressed for better readability.
   - Equation 5: \( e_i \) is not defined.
   - Figures 3 and 4 are not mentioned in the paper.
   - Table 2 is missing lines.

Key experimental details are missing. For example, the type of similarity reported in Figure 2 is not specified.

It is unclear whether the results in Table 5 are statistically significant. This information is crucial to validate the findings.

**Subjective Part Of Review:**

The paper would benefit greatly from providing the code and dataset via an open science platform such as an anonymous GitHub repository. This would facilitate the review process and enhance reproducibility.

The authors spend considerable effort explaining basic concepts like softmax and linear layer equations, which do not add significant value to the paper.

The reward function section lacks sufficient explanation. For example, the normalization and scaling of cosine similarities from 1 to 100 is not adequately justified.

The rationale behind the chosen research questions is not well-organized, and the repetition of experiments in sections 4.4 and 5 makes the paper repetitive.

---

### Meta-Review · Area_Chair_dkqU · 2024-05-24

**Recommendation:** Accept (Oral)
**Confidence:** 4

**Metareview:**

This is a meta-review. This paper addresses the problem of conflicts on social media. The authors propose a multi-objective classification model to identify both subtle forms and openly hostile forms of conflict. The reviewers all agree that the paper is of sufficient quality to be accepted, although they also list a number of weaknesses. Especially reviewer NFzY and WPug make some concrete comments that I suggest the authors to address in their revision. Overall, the paper addresses a relevant problem, and the methodology is sound. The annotated dataset is an important contribution and the authors make good suggestions for future work. The results are not all very convincing, but that is not a reason to reject the paper. (Note: The abstract has a placeholder not-functioning anonymous git URL; please replace by an actual URL in the revision.)